

# Adrenal SGLT1 or SGLT2 as predictors of atherosclerosis under chronic stress based on a computer algorithm

Jianyi Li[1,2,*], Lingbing Meng[1,2,*], Dishan Wu[1,2], Hongxuan Xu[1], Xing Hu[1], Gaifeng Hu[3], Yuhui Chen[4], Jiapei Xu[1], Tao Gong[4] and Deping Liu[1,2]

[1] Department of Cardiology, Beijing Hospital, National Center of Gerontology, Institute of Geriatric Medicine, Chinese Academy of Medical Sciences, Beijing, China

[2] Graduate School, Chinese Academy of Medical Sciences & Peking Union Medical College, Beijing, China

[3] Department of Cardiology, The First Affiliated Hospital of Wenzhou Medical University, Wenzhou, China

[4] Department of Neurology, Beijing Hospital, National Center of Gerontology, National Center of Gerontology, Institute of Geriatric Medicine, Chinese Academy of Medical Sciences, Beijing, China

[*] These authors contributed equally to this work.

## ABSTRACT

**Background.** Chronic stress promotes the development of atherosclerosis, causing disruptions in the body's hormone levels and changes in the structural function of organs.

**Objective.** The purpose of this study was to investigate the pathological changes in the adrenal gland in a model of atherosclerosis under chronic stress and to verify the expression levels of Sodium-glucose cotransporter (SGLT) 1 and SGLT2 in the adrenal gland and their significance in the changes of adrenal gland.

**Methods.** The model mice were constructed by chronic unpredictable stress, high-fat diet, and Apoe-/- knockout, and they were tested behaviorally at 0, 4, 8 and 12 weeks. The state of the abdominal artery was examined by ultrasound, and the pathological changes of the aorta and adrenal glands were observed by histological methods, and the expression levels and distribution of SGLT1 and SGLT2 in the adrenal gland were observed and analyzed by immunofluorescence and immunohistochemistry. The predictive value of SGLT1 and SGLT2 expression levels on intima-media thickness, internal diameter and adrenal abnormalities were verified by receiver operating characteristic (ROC) curves, support vector machine (SVM) and back-propagation (BP) neural network.

**Results.** The results showed that chronic stress mice had elevated expression levels of SGLT1 and SGLT2. The model mice developed thickening intima-media and smaller internal diameter in the aorta, and edema, reticular fiber rupture, increased adrenal glycogen content in the adrenal glands. More importantly, analysis of ROC, SVM and BP showed that SGLT1 and SGLT2 expression levels in the adrenal glands could predict the above changes in the aorta and were also sensitive and specific predictors of adrenal abnormalities.

**Conclusion.** SGLT1 and SGLT2 could be potential biomarkers of adrenal injury in atherosclerosis under chronic stress.

Corresponding authors
Deping Liu, lliudeping@263.net
Tao Gong, mac0852@163.com

## INTRODUCTION

With an aging population, the number of cardiovascular disease deaths in China has increased dramatically, with atherosclerotic cardiovascular disease (ASCVD) being the greatest burden. In recent years, the morbidity and mortality of atherosclerosis (AS) have been increasing significantly, and it has become a major underlying cause of various cardiovascular diseases (*Kobiyama & Ley, 2018*). The etiology and pathogenesis of AS are complex. AS is typically characterized by the formation of plaque at the intima of the arterial wall. The plaque consists of extracellular lipid particles and foam cells and their fragments, forming the lipid necrosis core. The core is surrounded by collagen-rich stroma and smooth muscle cells (SMCs), which are infiltrated by inflammatory cells that participate in plaque progression (*Libby et al., 2019*). Vascular senescence, especially that of vascular SMCs, endothelial cells and inflammatory cell, promotes plaque formation and increases the expression of inflammatory cytokines and chemokines (*Childs et al., 2016*).

With increasing competition in society and the deterioration of the ecological environment, the psychological and physical stress on human beings is increasing. A growing number of studies have noted that psychological stress such as depression and anxiety are important risk factors for ASCVD (*Cohen, Edmondson & Kronish, 2015*; *Rozanski, Blumenthal & Kaplan, 1999*). The INTERHEART study, conducted in more than 50 countries, showed that psychosocial stress, such as work stress and social isolation, was highly associated with cardiovascular endpoints (*Yusuf et al., 2004*). Other epidemiological studies have demonstrated that long-term stress and poor lifestyle were associated with an increased incidence of coronary heart disease and were strongly related to the development of AS (*Hintsanen et al., 2005a*; *Nabi et al., 2013*; *Osborne et al., 2020*). Stress is a potentially harmful stimulus that leads to psychological or physiological imbalance and nonspecific reactions in the body and can be divided into acute stress and chronic stress (*Han, Chen & Dong, 2015*). Acute stress resolves within a few days, and the symptoms disappear quickly, while chronic stress lasts longer and is more likely to cause adverse biological effects on the body (*McCarty, Horwatt & Konarska, 1988*). Chronic stress causes a sustained release of catecholamines (norepinephrine and epinephrine) and glucocorticoid hormone through the sympathetic adrenal medulla system and hypothalamic-pituitary-adrenal (HPA) cortical system, leading to neurohormonal disturbances in the body, which in turn promotes an inflammatory response, affects immune response, and endothelial vasodilatory function, and lead to the development of AS (*Meng et al., 2018*; *Inoue, 2014*). The adrenal glands can produce and secrete a variety of hormones and may be an important target for the treatment of chronic stress-related AS in the future. The pathological changes of the adrenal glands in AS under chronic stress and their molecular mechanisms are not clear; therefore, it is necessary to study the molecular mechanisms of adrenal gland changes in AS under chronic stress and their relationship with AS.

Sodium-glucose cotransporters (SGLTs) of a family of glucose cotransporters, mainly present in the small intestinal mucosa and proximal renal tubules, which use the difference in $Na^+$ concentration between the intracellular- and extracellular compartments to transport glucose and sodium ions into the cell (*Sano, Shinozaki & Ohta, 2020*). The

SGLT protein family comprises 12 members, SGLT1 to SGLT6 and 6 SLC5A proteins, of which SGLT1 and SGLT2 have been extensively studied because of their important physiological roles in glucose absorption in the small intestine and kidney, respectively (*Deng & Yan, 2016*). SGLT2 inhibitors show cardioprotective effects, but the mechanism of this protection is not yet clear (*Zelniker et al., 2019*; *Herat et al., 2020*). However, it is known that SGLT2 inhibitors can affect glucose metabolism and inhibit the sympathetic nervous system (*Gueguen et al., 2020*). The disturbance of glucose metabolism and alteration of the neuroendocrine axis may lead to changes in the adrenal glands. Therefore, from the point of view of the development of treatments for AS, it is important to clarify the relationship between the distribution and the expression levels of SGLT1 and SGLT2 in the adrenal gland, and adrenal pathological changes in chronic stress and the development of AS.

In this study, Apoe-/- mice were fed high-fat while administering chronic unpredictable mild stress (CUMS) to construct a stress-related AS model. The model was used to clarify the role of chronic stress in promoting AS and the status of the adrenal glands, as well as exploring the expression levels of SGLT1 and SGLT2 in the adrenal glands under stress. Finally, receiver operating characteristic (ROC) curves, support vector machine (SVM) and back-propagation (BP) neural networks were used to elucidate the relationship between SGLTs expression levels and the degree of abdominal atherosclerotic stenosis and pathological changes in the adrenal glands.

## MATERIALS & METHODS

### Animals and diets

Thirty male C57BL/6J mice and thirty male Apoe-/- mice were purchased from the Huafucang Biotechnology Co., Ltd. (Beijing, China) with animal license SCXK 2019-0008. All mice, in cages of five, were housed in a standard SPF environment, specifically $20-25$ °C room temperature, 40–70% humidity and free access to ultrafiltered water, with the exception of mice requiring chronic stress interventions that were on a 12h/12 h day and night cycle (*Hu et al., 2022*).

Sixty mice at 18–20 weeks, weighing 26–18 g, were randomly divided into four groups ($n = 15$ per group) using a random number table: (i) CON group: C57BL/6J mice were fed a normal diet for 12w; (ii) CON+CS group: C57BL/6J mice were fed a normal diet for 12w while receiving CUMS for 12w; (iii) HF+Apoe-/- group: Apoe-/- mice were fed high-fat for 12w; and (iv) HF+Apoe-/-+CS group: Apoe-/- mice were fed high-fat along with CUMS intervention for 12w. Fifteen mice per group to avoid decrease in statistical potency caused by attrition during the long-term chronic stimulations.

The experimental protocol was approved by Institutional Animal Care and Use Committee of Chinese Academy of Medical Sciences and Peking Union Medical College (CAMS&PUMC) with approval number LDP21001 and the approval date of July 28, 2021. All animal care and experiments were performed in strict followed by the Guide for the Care and Use of Laboratory Animals of the National Institutes of Health, which was always implemented in our previous studies (*Meng et al., 2021*; *Hu et al., 2022*). All mice died after cardiac blood collection under isoflurane anesthesia, and their aorta and adrenal glands were taken for subsequent testing.

## Construction and evaluation of the chronic stress model
### Chronic unpredictable mild stress

CUMS was used to construct a chronic stress model in mice and the detailed CUMS protocols has been described in our previous study (*Hu et al., 2022*). In brief, mice were subjected to different stress procedures daily for 12 weeks to avoid adaptation: tail capping for 1 min, sudden shaking for 5s each with an interval of 10s, day/night reversal, predation stress by putting mice in the same cage as rats, 70 dB noise stimulation, flash stimulation (150 flashes/min), bind to limit activity for 2 h, 45° cage tilt with turning it 180 degrees every hour for 24 h, and cage soiling for 24 h.

### Evaluation of chronic stress

Thirty minutes before behavioral assessment, the mice ($N = 60$) were placed in the test room and allowed to acclimatize to the new environment. The stress status of the mice in the four groups was evaluated at 0, 4, 8, and 12 weeks after the chronic stimulation as described previously (*Hu et al., 2022*). For the quiz, one mouse from the CON, CON+CS, HF+Apoe-/-, and HF+Apoe-/-+CS groups was placed in the device in turn until all mice were tested to exclude timing errors. Once the mouse was not in an anxious state after the assessment, then it would be shaved from the results.

### Elevated plus maze test

The elevated plus maze test (EPMT) was performed to evaluate the anxiety response of mice. Mice were placed in an open area for 5 min before being placed in the elevated maze to increase the activity of the mice in the maze. During the formal experiment, the mice were placed in the center area of the maze and then freely moved for 5 min. We observed and recorded the quiescent time of each group of mice at the total distance, open arm, closed arm and central area, respectively. Mice with shorter quiescent time at total distance and closed arm and longer quiescent time at open arm and central region indicated anxious state.

### Open field test

The open field test (OFT) is used to assess the autonomy and exploratory behavior of mice to new environments to reflect anxiety levels. Mice were placed in the center of the open field box ($50 \times 50 \times 40$ cm$^3$), which was divided into 16 small compartments, including 12 small compartments in the peripheral area and four small compartments in the central area. Mice were free to move around for 5 min. Video recording was performed to record the images and analyze the immobility time of mice in the total and non-central regions, as well as the number and duration of entry into the central area during the 5 min. After each experiment, the box was wiped with 75% alcohol to keep the box clean and odorless. A decrease in the number and duration of mice entering the central zone, a decrease in the total distance of quiescent time, and an increase in the non-central zone immobility time indicated that the mice were in an anxious state.

### Ultrasonic examination of the aorta

Mice ($N = 48$, 12 per group) were kept for 12 weeks and then underwent abdominal aorta ultrasound to assess the changes in the arterial wall. Mice were anesthetized with 4.5%

isoflurane (0.8-1L/min) according to the previous work (*Hu et al., 2022*), and the extent of AS was assessed by recording abdominal aortic intima-media thickness and internal diameter using an MS400 mouse electronic line array probe and an ultra-high frequency high-resolution small animal ultrasound imaging system (Vevo2100). Ultrasound was performed by a physician with animal ultrasonography experience at Peking University Third Hospital, and was unaware of the protocol and timing of the intervention.

## Histological examination

After assessing the stress level and vascular condition of the mice, mice were anesthetized with 4.5% isoflurane induced using a small animal anesthesia machine, followed by blood collection from cardiac puncture under 2% isoflurane maintenance anesthesia resulting in their death due to hemorrhage. All mice ($N = 60$) were euthanized to obtain their abdominal aorta and adrenal and embedded in Tissue-Tek OCT compound (Sakura, Tokyo, Japan) for sections or fixed in tissue fixation fluid followed by paraffin embedding, and subjected to hematoxylin-eosin (HE), periodic acid-schiff (PAS), silver staining, immunohistochemistry, and immunofluorescence staining.

## HE staining, PAS staining, and silver staining

The serial sections of the abdominal aorta and adrenal glands were stained with hematoxylin and eosin, and the intra-aortic atherosclerotic plaque area, lipid core size and the morphology of adrenal cells were observed under a microscope (Eclipse E100; Nikon, Tokyo, Japan). Adrenal paraffin specimens were dewaxed, and the reticular fibers were stained black with protein silver stain, and glycogen was stained purplish red with periodate and Schiff's solution, and the specimens were observed under different magnifications of the microscope.

## Immunofluorescence

Transverse sections of adrenal glands were incubated with SGLT1 Polyclonal Antibody (1:200, bs-1128R; BIOSS, Beijing, China) and SGLT2 Polyclonal Antibody (1:200, 24654-1-AP; Proteintech, Rosemont, IL, USA). Labeled SGLT1 in rose with Alexa Fluor 594-AffiniPure Goat Anti-Rabbit IgG (1:400; 111-585-003; Jackson ImmunoResearch, West Grove, PA, USA) and labeled SGLT2 in pink with Cy5 conjugated Goat Anti-Rabbit IgG (1:500; Servicebio, Wuhan, China). DAPI stain (G1012; Servicebio, Wuhan, China) was used to stain the nucleus in blue. In addition, CYP11B2 Polyclonal antibody (1:200, 20968-1-AP; Proteintech, Rosemont, IL, USA) and Cy3 conjugated Goat Anti-Rabbit IgG (1:500; Servicebio, Wuhan, China) was used to label the adrenal cortex in red, and tyrosine hydroxylase (TH) Polyclonal antibody (1:100, 25859-1-AP; Thermo Fisher Scientific, Massachusetts, USA) and Alexa Fluor 488-Conjugated AffiniPure Goat Anti-Rabbit IgG (1:200, A-11008; Thermo Fisher Scientific, Waltham, MA, USA) was used to demarcate the adrenal medulla in green to further characterize the expression regions of SGLT1 and SGLT2. The distribution of SGLT1 and SGLT2 in adrenal tissues at different magnifications was observed by fluorescence microscope (Eclipse E100; Nikon, Tokyo, Japan), and the expression levels of the two proteins were assessed by Image-Pro Plus 6.0 software (Media Cybernetics, Inc., Rockville, MD, USA).
### Immunohistochemistry

Paraffin sections of the adrenal glands were stained with SGLT1 antibody (1:200, BIOSS, Beijing, China) and SGLT2 antibody (1:200, Proteintech, Rosemont, USA), and incubated with horseradish peroxidase (HRP) secondary antibodies. The target proteins were localized by a color development reaction using a freshly prepared DAB chromogenic solution which stained the target proteins in brownish yellow. The nuclei were re-stained blue by hematoxylin staining solution. The staining results were observed under a light microscope (Eclipse E100; Nikon, Tokyo, Japan), and at least six randomly selected fields of view at 20x magnification from each section were analyzed. The quantitative expression of SGLT1 and SGLT2 in the adrenal gland was analyzed using Image-Pro Plus 6.0 software (Media Cybernetics, Inc., Rockville, MD, USA) by calculating the integral optical density (IOD) / area of interest (AOI) of the positively stained region.

### Western Blot

Protein was extracted from adrenal tissue for Western blot. The protein lysates were separated in 10% SDS-PAGE and transferred to PVDF membrane. PVDF membrane was closed at room temperature with 5% skimmed milk dissolved in Tris-buffered saline with Tween-20 (TBST) for 1 h, followed by incubation of the primary antibody in a 4 °C overnight. The next day, after washing the membrane with TBST for 5 min $\times 3$, the membrane was incubated with secondary antibody for 1 h at room temperature. After the incubation was completed, the membranes were washed again 3 times and developed by ECL method. The primary antibodies and dilution ratios used in this study were as follows: anti-SGLT2 (1:1000, ab14686, abcam), anti-SGLT2 (1:1000, ab37296, abcam), GAPDH (1:5000, 60004-1-Ig, proteintech).

### Quantitative real-time polymerase chain reaction (qPCR)

To clarify the mRNA levels of adrenal SGLT2. Total RNA was collected from mouse adrenal tissue using TRIzol (Takara, 9108) and reverse transcribed to cDNA. The quantification of real time PCR using QuantiTect SYBR Green master mix (Qiagen). The $2^{-\Delta\Delta Ct}$ was used to calculate relative gene expression levels. The primer sequences for SGLT1: 5′-AATGCGGCTGACATCTCAGTC-3′ (Forward), 5′-ACCAAGGCGTTCCATTCAAAG-3′ (Reverse); SGLT2: 5′-GAGCAACACGTAGAGGCAGG-3′ (Forward), 5′-GCAGCGATAACCAGAATGTCA-3′ (Reverse); GAPDH: 5′-GGTCCCAGCTTAGGTTCATCAGGT-3′ (Forward), 5′-AATACGGCCAAATCCGTTCACACC-3′ (Reverse).

### Support vector machine

SVM was used to process the data to construct a binary functional model for data analysis. The two input variables were the independent variables, namely SGLT1 and SGLT2 expression levels in the adrenal gland, and the output variables were adrenal gland edema, broken reticular fiber, glycogen content, intima-media thickness, and internal diameter, which were the dependent variables.

We trained the network using a regularization adjustment method that adjusted the performance function of the network as described in the previous study (*Hu et al., 2022*). Regularization is a form of regression that bounds, adjusts, or shrinks the coefficient

estimate toward zero. In other words, regularization reduces the model complexity and instability during the learning process, thus avoiding the danger of overfitting. By regularizing and adjusting the performance function, improving the generalization performance of the network.

## Construction of BP -neural network model

BP-neural network is a kind of multi-layer feedforward network trained by error back propagation. The signal is first forward propagated according to the input layer-hidden layer-output layer, then backpropagation in this reverse order, and adjusts the weights and biases from the hidden layer to the output layer and from the input layer to the hidden layer in turn.

In this study, the normalization processing of variable values, network simulation, network training, and network initialization was done through MATLAB (version 2014a).

The two input variables were the independent variables described above. The output variables are the adrenal-related dependent variables described above. Then the original data was divided into 70% of the test set for training to build a predictive model and 30% of the test set, and the trained predictive model was applied to the test set for prediction. Finally, we used a cubic spline interpolation algorithm to analyze the high-risk warning of dependent variables content on independent variables as described in the previous work (*Hu et al., 2022*).

## Statistics

The SPSS software, version 24.0 (IBM Corp., Armonk, NY, USA) was used for statistical analysis. Continuous variables were presented as mean $\pm$ Standard error of mean (SEM). Unpaired Student's $t$-test was applied for comparison between two groups. Correlations comparing SGLT1, SGLT2 expression levels, adrenal edema, broken reticular fibers, glycogen content, intima-media thickness, and internal diameter were analyzed using Spearman's and Pears-rho tests, and the sensitivity and specificity of SGLT1 and SGLT2 expression levels for predicting adrenal gland lesions were analyzed by ROC. The mean of different adrenal damages in the mice was calculated, and threshold was set. Mice with damage greater than this threshold were assigned a value of 1, and those with less than this threshold were assigned a value of 0. There was a statistical difference when $P \ll 0.05$; *$P < 0.05$, **$P < 0.01$, *** $P < 0.001$; ns =no significance.

## RESULTS

### Evaluation of mouse model for atherosclerosis under chronic stress
*Evaluation of chronic stress*

The EPMT and OFT were performed to assess the stress status of mice after 0, 4, 8, and 12 weeks of chronic unpredictable stress. All indicators of EMPT (quiescent time in different areas of the maze), were not significantly different among the four groups at baseline (Fig. 1A). After four weeks of CUMS, the quiescent time in the total distance of mice in the HF+Apoe-/-+CS group was lower than that of the CON group, and the quiescent time in the closed arms of the CON+CS group was significantly decreased compared with that

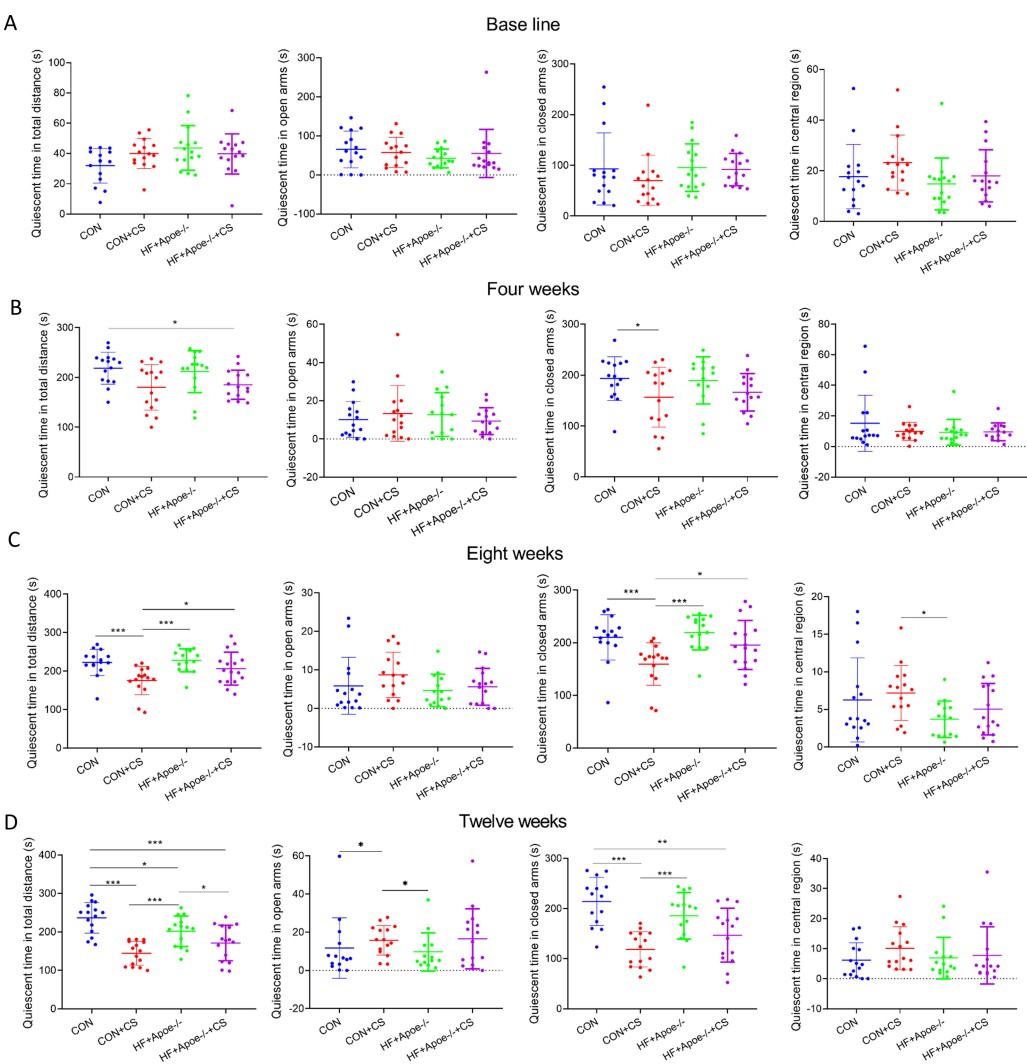

**Figure 1   Behavioral results of elevated plus maze.** (A) At baseline, there were no significant differences in the quiescent time in the distance, open arm, closed arm, and the central region among the four groups of mice. (B–D) Changes in the quiescent time in the above four regions between the four groups after 4, 8 and 12 weeks of CUMS intervention, respectively, reflecting the degree of excitation in mice. * $P < 0.05$, ** $P < 0.01$, *** $P < 0.001$; ns =no significance.

of the CON group (Fig. 1B). After eight weeks of CUMS, quiescent time in total distance and quiescent time in closed arms were significantly decreased in the CON+CS group mice compared with the CON group and HF+Apoe-/- group (Fig. 1C). After 12 weeks of CUMS the CON+CS group differed from the CON group in terms of the quiescent time in total distance, open arm and closed arm. The HF+Apoe-/-+CS group had statistical differences in quiescent time in total distance from the CON group and HF+Apoe-/- group (Fig. 1D).

In the OFT, there was no differences in total quiescent time, number of times entering the centrals, duration in the centrals and non-center immobility time between the four groups (Fig. 2A). After four weeks of CUMS, mice in both stress groups had less total quiescent

time than the two non-stressed groups, but the number of entries into the centrals increased in the CON+CS and HF+Apoe-/-+CS groups compared to the HF+Apoe-/- group (Fig. 2B). After eight weeks, the total quiescent time in the CON+CS group was less than that in the CON group, and the total quiescent time in the HF+Apoe-/-+CS group was less than that in the HF+Apoe-/- group (Fig. 2C). After twelve weeks, the total quiescent time in the CON was greater than that in the CON+CS group, and the mice in the CON+CS and HF+Apoe-/-+CS groups entered the central area less frequently than those in the CON and HF+Apoe-/- groups, and the former also had a shorter duration in the centrals and a corresponding increase in quiescent time in the non-central area (Fig. 2D). Overall, the stressed mice showed states of anxiety and depression in most of the indicators of EPMT and OFT.

*Evaluation of atherosclerosis*

Twelve mice in each group were randomly selected for abdominal aortic ultrasonography to visualize the extent of arterial lesions. The results showed that the intima of the abdominal aorta in HF+Apoe-/-+CS mice was poorly defined, rough, with interrupted continuity and thickened walls, and the typical plaques protruding from the lumen (Fig. 3A). Moreover, compared with CON, CON+CS and HF+Apoe-/- groups, HF+Apoe-/-+CS mice had increased intima-media thickness and decreased diameter of the abdominal aorta, indicating that early atherosclerotic changes were clearly observed in the model mice. In addition, mice in the CON+CS group had thickened intima-media and smaller diameters of the abdominal aorta compared with the CON group (Figs. 3B–3C).

HE staining of the aorta revealed a clear presence of plaques in the intima of the mice artery in the HF+Apoe-/- and HF+Apoe-/-+CS groups, and the plaque size was larger in the HF+Apoe-/-+CS group than in the HF+Apoe-/- group. However, in the CON and CON+CS groups, there was no significant plaque in the intima (Fig. 3D).

## Effects on the adrenal gland in atherosclerosis under chronic stress

The area of adrenal edema was significantly increased in the mice of the HF+Apoe-/-+CS group compared with the CON and HF+Apoe-/- groups, suggesting that AS and chronic stress may have accelerated adrenal edema (Fig. 4A). Greater degree of broken reticular fibers in the adrenal glands of the HF+Apoe-/-+CS group, whereas there was no difference in reticular fiber breakage in the CON+CS group compared to the CON group (Fig. 4B). The proportion of glycogen stained purplish red in the adrenal glands was significantly increased in both the HF+Apoe-/-+CS group and the CON+CS group compared with the HF+Apoe-/- group and CON group, respectively (Fig. 4C), which may be related to chronic stress promoting sugar reabsorption. The above result data were the mean ±SEM. The Spearman correlation analysis showed a closely positive correlation between the degree of reticular fiber breakage and the edema area of the adrenal gland ($R = 0.842$, $P < 0.001$), and the glycogen content in the adrenal gland was strongly related to the edema area of the adrenal gland ($R = 0.933$, $P < 0.001$). Additionally, the proportion of glycogen was also related to the degree of broken reticular fibers ($R = 0.777$, $P < 0.001$) (Fig. 4D).

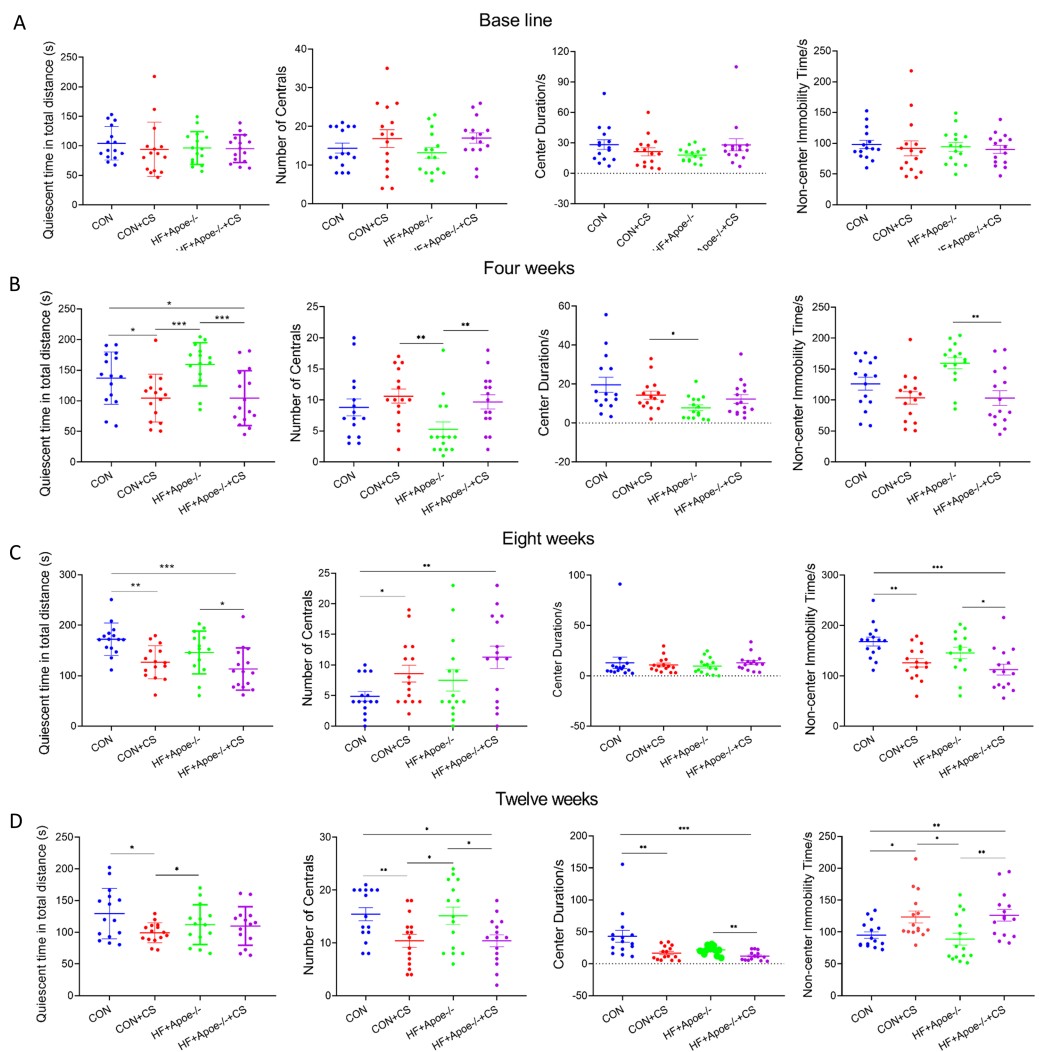

**Figure 2   Behavioral results of the open-field test.** (A–D) Changes in the quiescent time in distance, number of times entering the centrals, duration in the center and non-center immobility time in four groups at baseline, and after 4, 8 and 12 weeks of CUMS intervention. * $P < 0.05$, ** $P < 0.01$, *** $P < 0.001$; ns =no significance.

## Expression levels and distribution of SGLT1 and SGLT2 in the adrenal gland

### Results of Western blot qPCR and immunofluorescence

Figures 5A–5B showed typical samples of SGLT1 and SGLT2 staining in the adrenal tissue, where SGLT1 was stained in a rose color and SGLT2 in pink. After semiquantitative analysis, SGLT1 expression was upregulated in the adrenal glands of the CON+CS group and HF+Apoe-/-+CS group compared with the CON and HF+Apoe-/- groups, in which the expression of SGLT1 in the adrenal glands of mice in the HF+Apoe-/-+CS group was higher than the other three groups. Similarly, SGLT2 expression levels in the adrenal glands in the HF+Apoe-/-+CS group and CON+CS group were higher than those in the CON

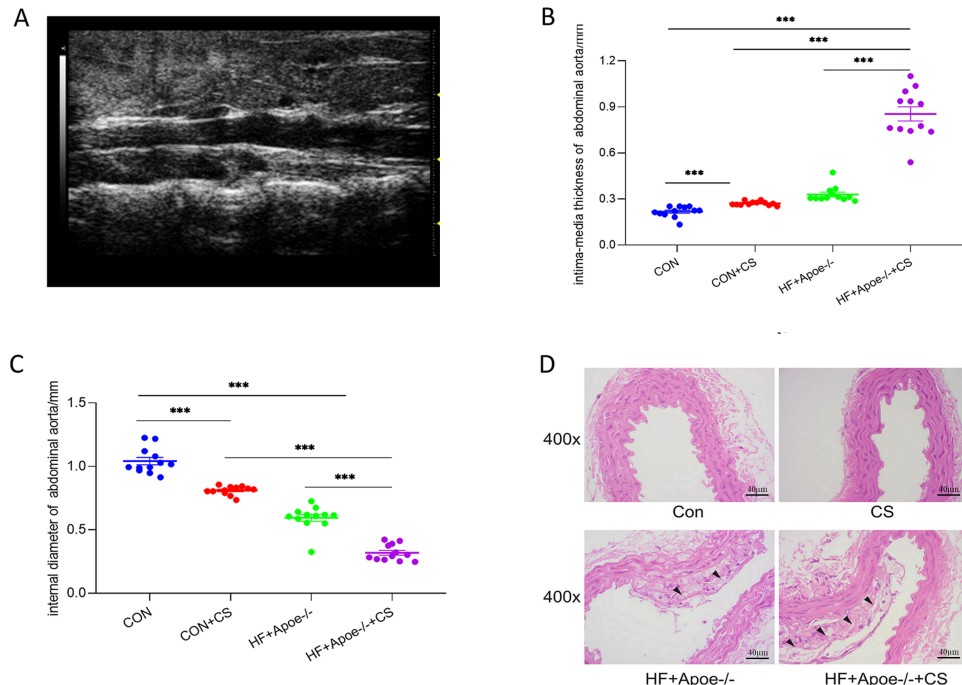

**Figure 3** **Atherosclerosis model evaluation.** (A) Typical irregular plaque formation and luminal stenosis with disrupted continuity in the abdominal aorta of HF+Apoe-/-+CS mice. (B) The intima-media thickness of the abdominal aorta in different groups of mice. (C) Intra-aortic diameter in different groups of mice. (D) HE staining of the aorta of different groups of mice showing plaque formation in HF+Apoe-/- and HF+Apoe-/-+CS groups.

group but the SGLT2 expression level in the HF+Apoe-/-+CS group was the highest among the four groups (Fig. 5C). The above result data were the mean ±SEM.

Western and qPCR analysis showed that the protein and mRNA expression levels of adrenal SGLT1 and SGLT2 were significantly increased after chronic stress, specifically, the levels of SGLT1 and SGLT1 in the CON+CS group were higher in the CON group and the levels of SGLT1 and SGLT2 in the HF+Apoe-/-+ CS group were higher than those in the HF+Apoe-/- group (Figs. 5D–5E).

The adrenal cortex and medulla were differentiated with fluorescent anti-CYP11B2 and anti-TH, respectively, and the results of immunofluorescence showed that SGLT1 was mainly expressed in the medullary region and a small amount in the cortex region in the absence of chronic stress intervention. The expression of SGLT1 in the cortex region increased after chronic stress intervention. SGLT2 was mainly distributed in the adrenal medulla with or without chronic stress intervention (Fig. 6).

## Results of immunohistochemistry

Immunohistochemical staining showed that SGLT1 expression was upregulated in the adrenal gland of the CUMS-intervened mice than in non-intervened mice (Fig. 7A). Compared with the CON group, SGLT2 expression in the adrenal glands was higher in the other three groups, specifically, the HF+Apoe/-+CS group had the highest SGLT2

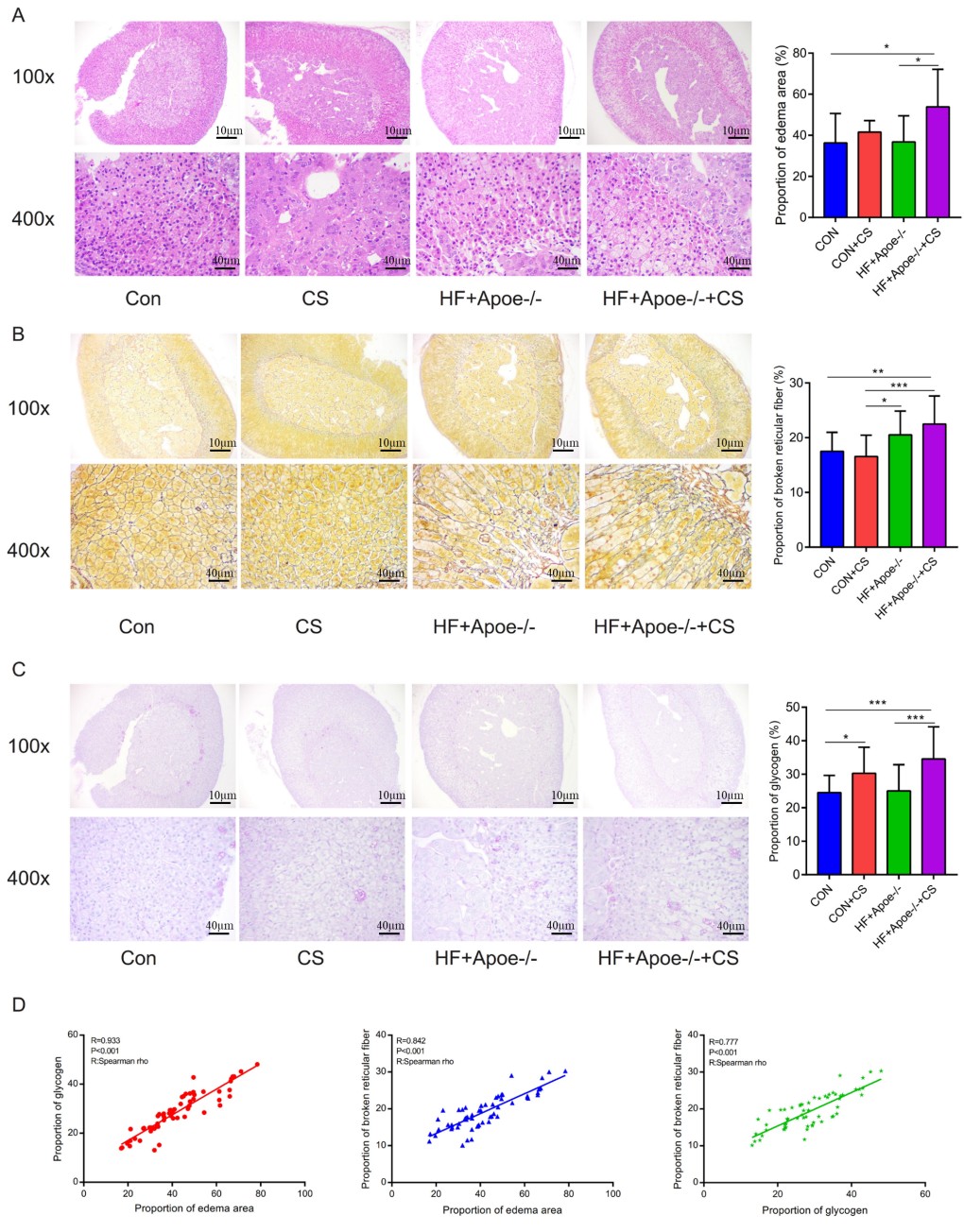

**Figure 4 Atherosclerosis under chronic stress leaded to adrenal injury.** (A) HE staining showed different degrees of adrenal edema in the four groups. (B) Silver staining demonstrated that black-stained reticular fibers were broken significantly in the HF+Apoe-/- and HF+Apoe-/-+CS groups. (C) Glycogen staining revealed higher glycogen content in the CON+CS group and HF+Apoe-/-+CS group than that in the CON group and the HF+Apoe-/- group. (D) There was a positive correlation between glycogen ratio, reticular fiber breaks, and adrenal edema area. * $P < 0.05$, ** $P < 0.01$, *** $P < 0.001$; ns =no significance.

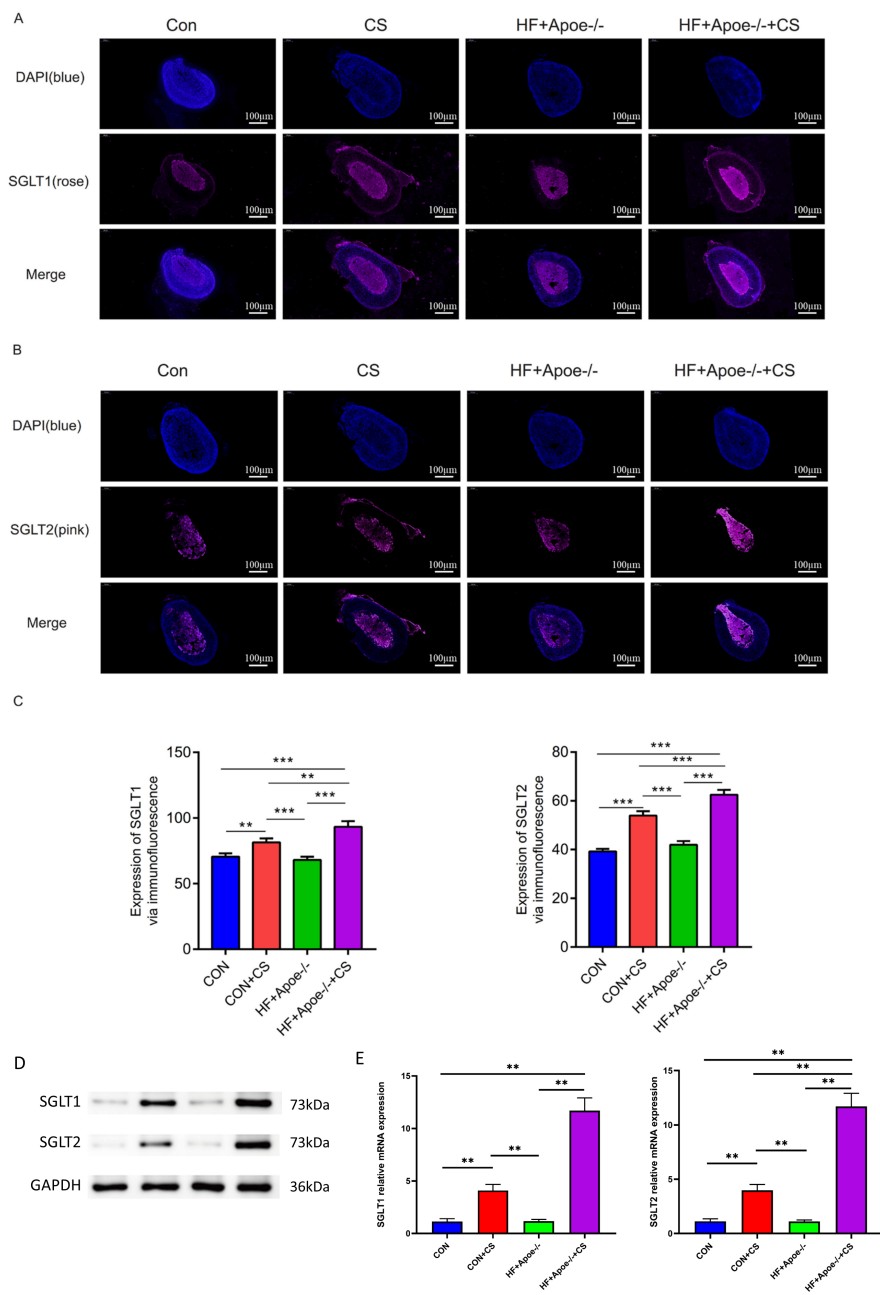

**Figure 5  SGLT1 and SGLT2 expression levels in the adrenal gland.** (A–B) Rose-colored SGLT1 fluorescence and pink-colored SGLT2 fluorescence could be seen in the adrenal cross sections of the four groups. (C) Semi-quantitative analysis of SGLT1 and SGLT2 expression in the adrenal glands of four groups showed that SGLT1 and SGLT2 expression levels were higher in mice after stress than in the corresponding non-stressed mice. (D) Expression of SGLT1 and SGLT2 was determined by Western blot analysis. (E) mRNA expression of SGLT1 and SGLT2 was detected by qPCR.

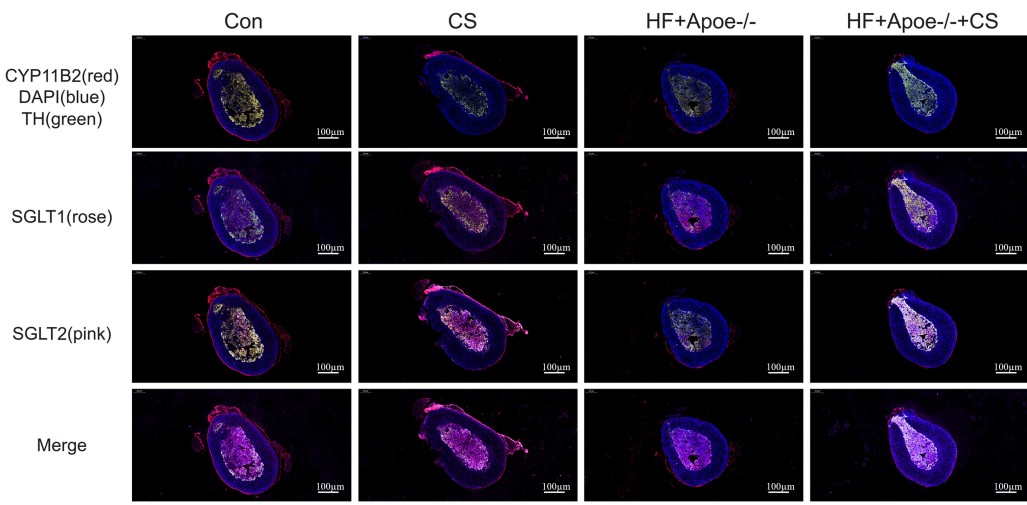

**Figure 6 Distribution of SGLT1 and SGLT2 in the adrenal gland.** CYP11B2 labeled in red showed the adrenal cortex region, while tyrosine hydroxylase (TH) labeled in green showed the adrenal medulla. The nucleus was stained blue by DAPI. SGLT1 was detected by a rose and SGLT2 by a pink. The distribution of SGLT1 and SGLT2 in the adrenal glands of mice in the CON, CON+CS, HF+Apoe$^{-/-}$, and HF+Apoe$^{-/-}$+CS groups was observed to be altered by chronic stress.

expression in the adrenal glands, which was consistent with the immunofluorescence results (Fig. 7B). The above result data were the mean ±SEM.

Spearman correlation analysis showed that the expression levels of SGLT1 and SGLT2 in the adrenal gland were associated with the degree of edema area ($R = 0.678$, $P < 0.001$; $R = 0.700$ $P < 0.001$), broken reticular fiber ($R = 0.657$, $P < 0.001$; $R = 0.570$ $P < 0.001$), and abnormal glycogen content ($R = 0.748$, $P < 0.001$ $R = 0.958$ $P < 0.001$), respectively. The heat map further showed the positive correlations between the five indicators of edema, glycogen, reticular fiber, adrenal SGLT1 and SGLT2 (Fig. 7C).

## Regression prediction of intima-media thickness and internal diameter in the abdominal aorta by BP neural network

After training of BP-neural network for using SGLT1 and SGLT2 expression levels for predicting intima-media thickness, the best training performance was 0.0096775 at epoch 3000 (Fig. 8A). That means the network error was 0.0097 after 3000 times of network training, which reached the requirement of the network error setting. Meanwhile, the MSE was gradually stabilized and the training set area became straight as the number of training times increased. The relativity of training was 0.9841, which means that the goodness of fit of the training sample was 98.41%, indicating good fitness of the network training model, and that the model can be fully utilized for the prediction of the intima-media thickness of the abdominal aorta (Fig. 8B). Using the trained BP neural network to verify the predicted data with the original values, we found that there was gradually no significant error between the two as the sample size increased (Figs. 8C–8D). In summary, the intima-media thickness of the abdominal aorta correlated with the expression levels of SGLT1 and SGLT2 in the adrenal glands, and the latter could be used to predict the degree of change in the former.

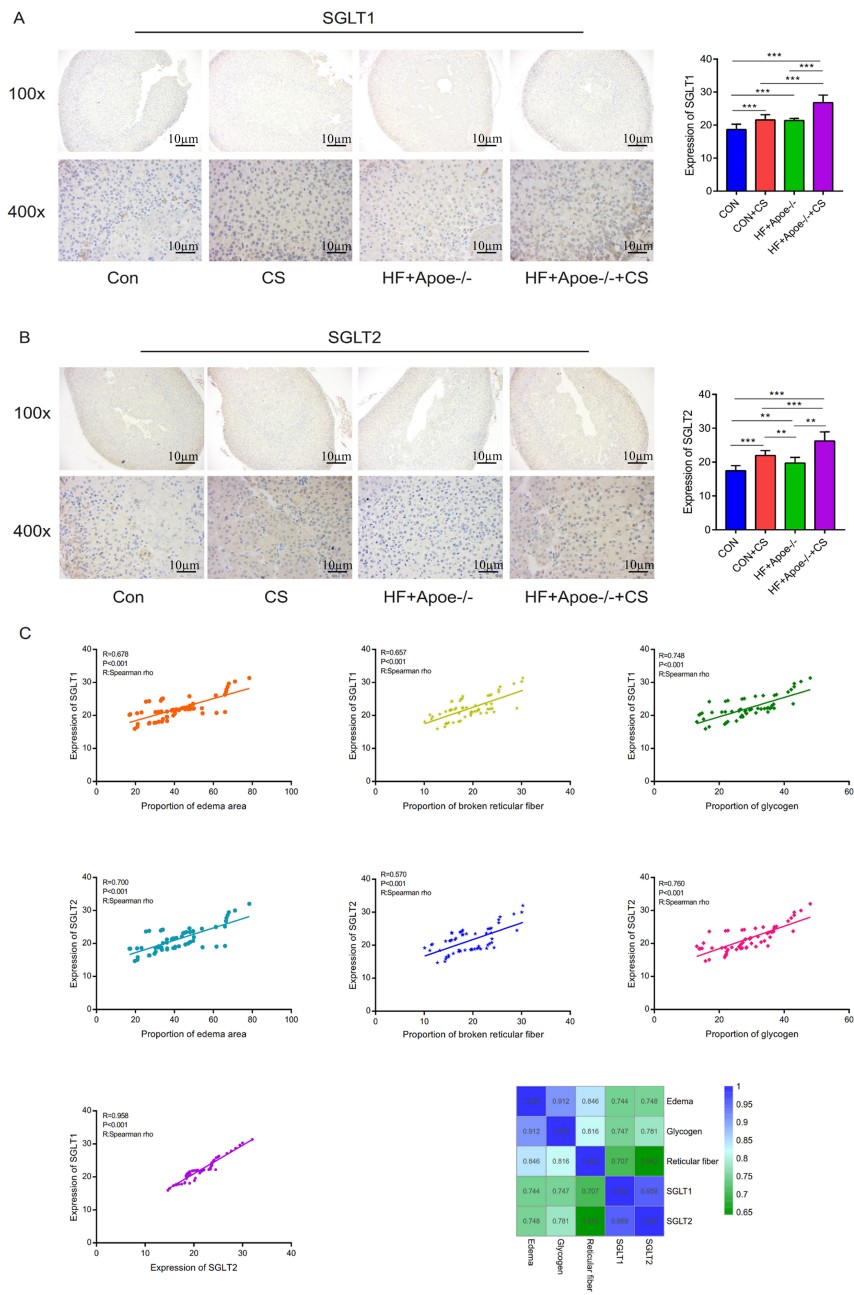

**Figure 7 Distribution and significance of SGLT1 and SGLT2 in the adrenal gland based on immuno-histochemistry.** (A) The distribution of SGLT1 was observed under low and high magnification, and the differences in SGLT1 expression level in the adrenal gland of mice in the four groups were analyzed by semi-quantitative methods. (B) The distribution of SGLT2 was observed under high and low magnification in the adrenal gland, and the differences in expression between the four groups were analyzed. (C) Edema, glycogen content, and reticulocyte fiber breakage were positively correlated with the expression levels of SGLT1 and SGLT2 in the adrenal gland, respectively.\* $P < 0.05$, \*\* $P < 0.01$, \*\*\* $P < 0.001$; ns =no significance.

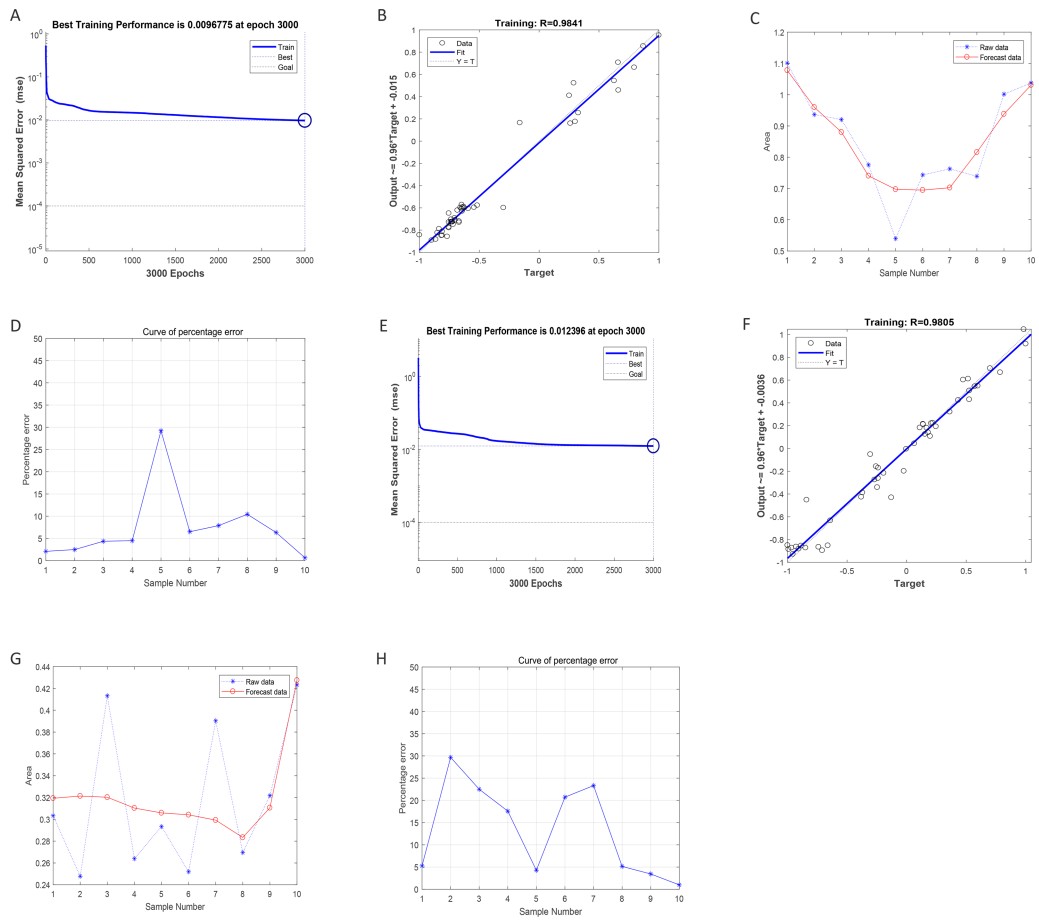

**Figure 8 Neural network models prediction of the intima-media thickness and internal diameter of the abdominal aorta.** (A) Best training score of 0.0096775 with an epoch of 3000 after BP-neural network training for predicting intima-media thickness of abdominal aorta from SGLT1 and SGLT2 expression levels. (B) Relativity of 0.9841 with good correlation between input and output quantities. (C–D) Validation of the predicted data and the original values, with only small differences. (E) Best training score of 0.012396 with an epoch of 3000 after training the BP-neural network for predicting the internal diameter of abdominal aorta from SGLT1 and SGLT2 expression levels. (F) Relativity of 0.9805, with a good correlation between input and output volumes. (G–H) Validation of predicted data and original values.

Similarly, the trained BP neural network had the best score of 0.012396 at epoch 3000 with relativity of 0.9805 (Figs. 8E–8F). Validation of the predicted data with the original values, showed some error between the two; still, the model fitted the internal diameter of aorta well when the sample size was large enough, predicting the change in internal diameter with the expression levels of SGLT1 and SGLT2 (Figs. 8G–8H).

## The sensitivity and specificity of adrenal SGLT1 and SGLT2 expression in the diagnosis of adrenal injury

The ROC curves fitted for SGLT1/2 and glycogen content abnormalities were close to the upper left corner of the coordinate axis, indicating that the sensitivity and specificity of using the former to predict the latter was high, and in terms of specific values, the area

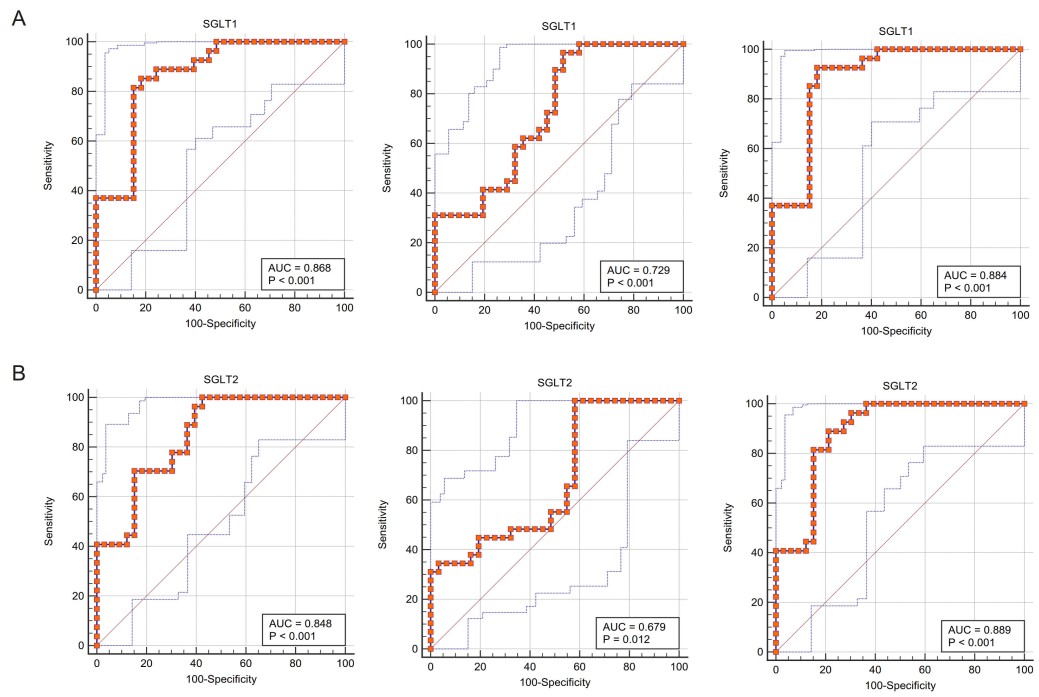

**Figure 9  ROC curves validated the predictive value of SGLT1 and SGLT2 for the degree of adrenal injury.** (A) The ROC curves showed that the expression level of SGLT1 in the adrenal gland sensitively and specifically predicted adrenal edema, broken reticular fibers, and glycogen content. (B) The ROC curves showed that the expression level of SGLT2 in the adrenal gland sensitively and specifically predicted adrenal edema, broken reticular fibers, and glycogen content.

under curves (AUCs) were all greater than 0.85, indicating that SGLT1/2 was of great value to predict glycogen content abnormalities. Similarly, the sensitivity and specificity of SGLT1/2 for predicting adrenal edema were also high (AUC 0.868, $P < 0.001$; 0848; $P < 0.001$), respectively, and the value of predicting reticular fiber breaks was also good (AUC 0.729, $P < 0.001$; 0.679, P 0.05, respectively) (Figs. 9A–9B).

## Predictive significance of SGLT1 and SGLT2 expression levels on edema, reticular fiber and glycogen in the adrenal gland *via* SVM

SVM was used to impute the predictive value of SGLT1 and SGLT2 expression levels to correspond to the variables, and the incidence was imputed by comparison of predicted and actual values. After the calculation of SVM, the predictive value of SGLT1 and SGLT2 expression levels for the adrenal gland edema was 0.9596 (y = 0.7706*x+10.9511) with the mean error of 3.31%, 0.8246 (y = 0.6508*x+7.1683) for broken reticular fibers with the mean error of 1.81%, and 0.9478 (y = 0.6987*x+10.1828) with the mean error of 2.15% for glycogen content (Figs. 10A–10C). Overall, the predictive value of SGLT1/2 for the three types of adrenal gland injuries was high, and the fit between the predicted and actual values increased with the increase in sample number.

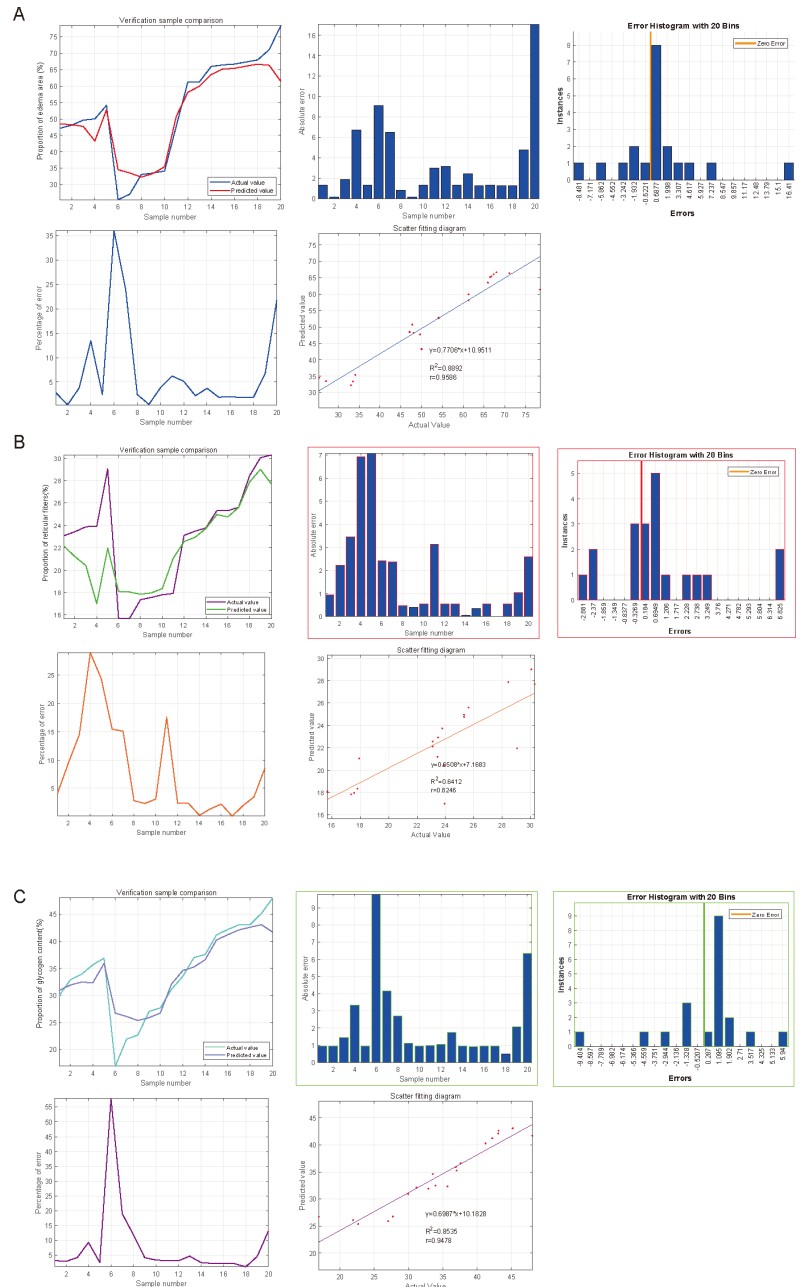

**Figure 10 Predicted significances of SGLT1 and SGLT2 expression levels on adrenal edema, reticulocytes, and glycogen.** (A) The predicted value of SGLT1 and SGLT2 expression levels on adrenal edema was 0.9596 with a mean error of 3.31% by the SVM method. (B) The predicted value of SGLT1/2 expression levels for broken reticular fibers was 0.8246 with a mean error of 1.81%. (C) The predicted value of SGLT1/2 expression levels for glycogen content was 0.9478 with a mean error of 2.15%.

### Prediction and high-risk warning range of adrenal edema, reticular fiber breakage, and abnormal glycogen content based on BP-neural network

The best training performance of the BP-neural network for predicting adrenal edema from SGLT1 and SGLT2 expression levels was 0.018092 at epoch 2999, with relativity of 0.96057 (Figs. 11A–11B). By verifying the predicted data with the original values, only small differences were found (Figs. 11C–11D). Based on these results, we speculated that SGLT1 and SGLT2 expression might be a predictor of adrenal edema. We then determined the high-risk warning indicators of adrenal edema by cubic spline interpolation, the risk of adrenal edema increased sharply when the SGLT1 expression level was 16–25%, and SGLT2 expression level was greater than 28% (Fig. 11E). In addition, we build a three-dimensional (3D) stereogram to clarify the warning range well (Fig. 11F).

After training of BP neural network for adrenal reticular fiber breakage from the expression levels of SGLT1 and SGLT2, the best training performance was 0.01571 at epoch 3000 times and the relativity was 0.96383 (Figs. 12A–12B). There was no significant difference between the forecast data and the raw data (Figs. 12C–12B). Based on the above, we could speculate that SGLT1 and SGLT2 expression might be a predictor of adrenal reticular fiber breakage. Figure 12E showed a map of high-risk warning indicators for adrenal reticular fiber breakage derived from SGLT1 and SGLT2 expression levels. The risk of adrenal reticular fiber rupture was significantly increased when the SGLT1 and SGLT2 expression levels were 16–28% and 28%, respectively. In addition, the 3D map presented in Fig. 12F presented a good warning range.

The best training performance for the prediction of abnormal glycogen content in from SGLT1 and SGLT2 expression levels in the adrenal gland was 0.013784 (Fig. 13A), and the relativity was 0.97073 (Fig. 13B). There was only a small difference between the forecast data and the raw data (Figs. 13C–13D). High-risk warning indicators for adrenal glycogen content were determined (Fig. 13E). SGLT1 expression level in the range of 16–31%, and SGLT2 expression levels in the range greater than 30% were very likely to occur with abnormal adrenal glycogen content. In addition, SGLT1 expression level of 21–28%, and SGLT2 expression levels of 25–30% indicated the possibility of adrenal gland injury. The 3D stereogram gives a good representation of the warning range (Fig. 13F).

## DISCUSSION

In this study, we found that the expression of both adrenal SGLT1 and SGLT2 were increased under chronic stress in mice. Atherosclerotic mice subjected to chronic stress showed increased intima-media thickness and decreased internal diameter of the abdominal aorta, edema, reticular fiber disruption, and abnormal glycogen content in the adrenal gland, and these alterations were associated with elevated levels of SGLT1 and SGLT2 in the adrenal gland. SGLT1 and SGLT2 could predict the extent of atherosclerosis and adrenal injury in Apoe-/- mice under chronic stress, as elucidated by both SMV and BP neural network models.

Since the 21st century, cardiovascular diseases have become the number one killer, and AS is the most important cause of cardiovascular disease, which can lead to common

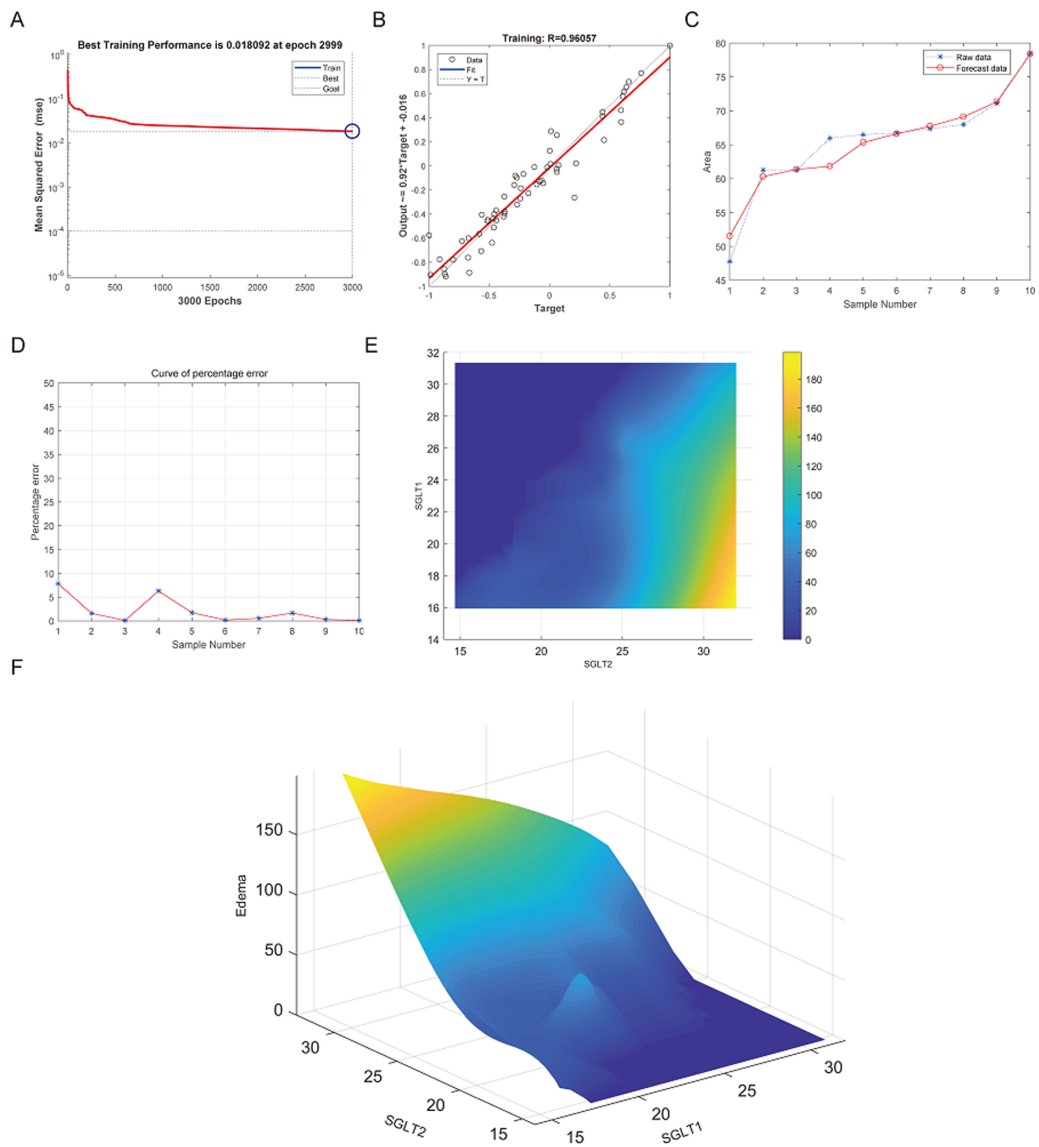

**Figure 11 Prediction and high-risk warning range of adrenal edema based by SGLT1 and SGLT2 expression levels.** (A–B) Best training score of 0.018092 at epoch 2999 and relativity of 0.96057 of BP-neural network for predicting adrenal edema from SGLT1 and SGLT2 expression levels after training. (C–D) Validation of the predicted values with the original values revealed only small differences. (E–F) A high-risk warning indicator for adrenal edema derived from SGLT1 and SGLT2 expression levels was found by an interpolation algorithm and presented with a three-dimensional (3D) stereogram of the warning range.

diseases such as myocardial infarction and coronary heart disease (*Leong et al., 2017*). AS is characterized by progressive lipid deposition, fibroplasia, and inflammatory cell infiltration in the arterial plaques, and AS is related to inflammation, immune response, and autophagy (*Shao et al., 2016*; *Wolf & Ley, 2019*). Chronic stress is an important predisposing factor for many psychological and physical disorders, and studied have shown that some patients with AS who do not have traditionally defined risk factors develop the disease

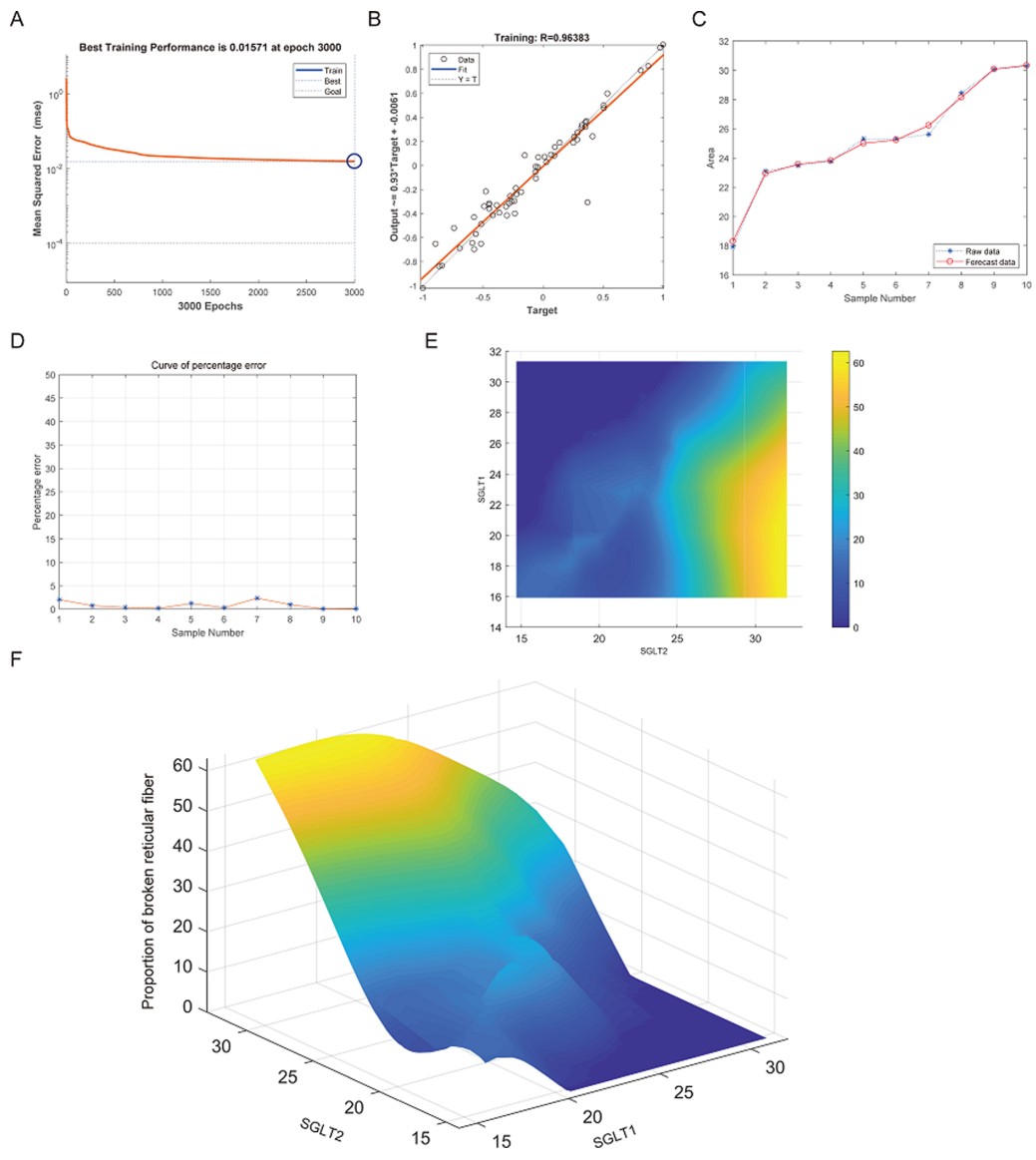

**Figure 12** **Prediction and high-risk warning range of adrenal reticular fiber breakage based by SGLT1 and SGLT2 expression levels.** (A–B) The optimal training score for SGLT1 and SGLT2 to predict adrenal reticular fiber breakage was 0.01571 at epoch 3000 with relativity of 0.96383. (C–D) The predicted data were verified against the original values and a significant difference was found between the two. (E–F) High-risk warning indicators for adrenal reticular fiber breakage were determined from SGLT1 and SGLT2 expression levels, and the warning range was presented with a 3D stereogram.

due to chronic stress (*Rudisch & Nemeroff, 2003*). Chronic stress is generally defined as a subjective perception made in response to long-term persistent or repeated adverse external stimuli, which caused by sustained internal arousal, usually accompanied by neurological, endocrine, immunological and behavioral changes to adapt to the new situation (*Cohen et al., 2012*; *Bangasser & Sanchez, 2020*). It has been shown that stress responses, including stimulation of the sympathetic nervous system (SNS) and HPA cortical system associated

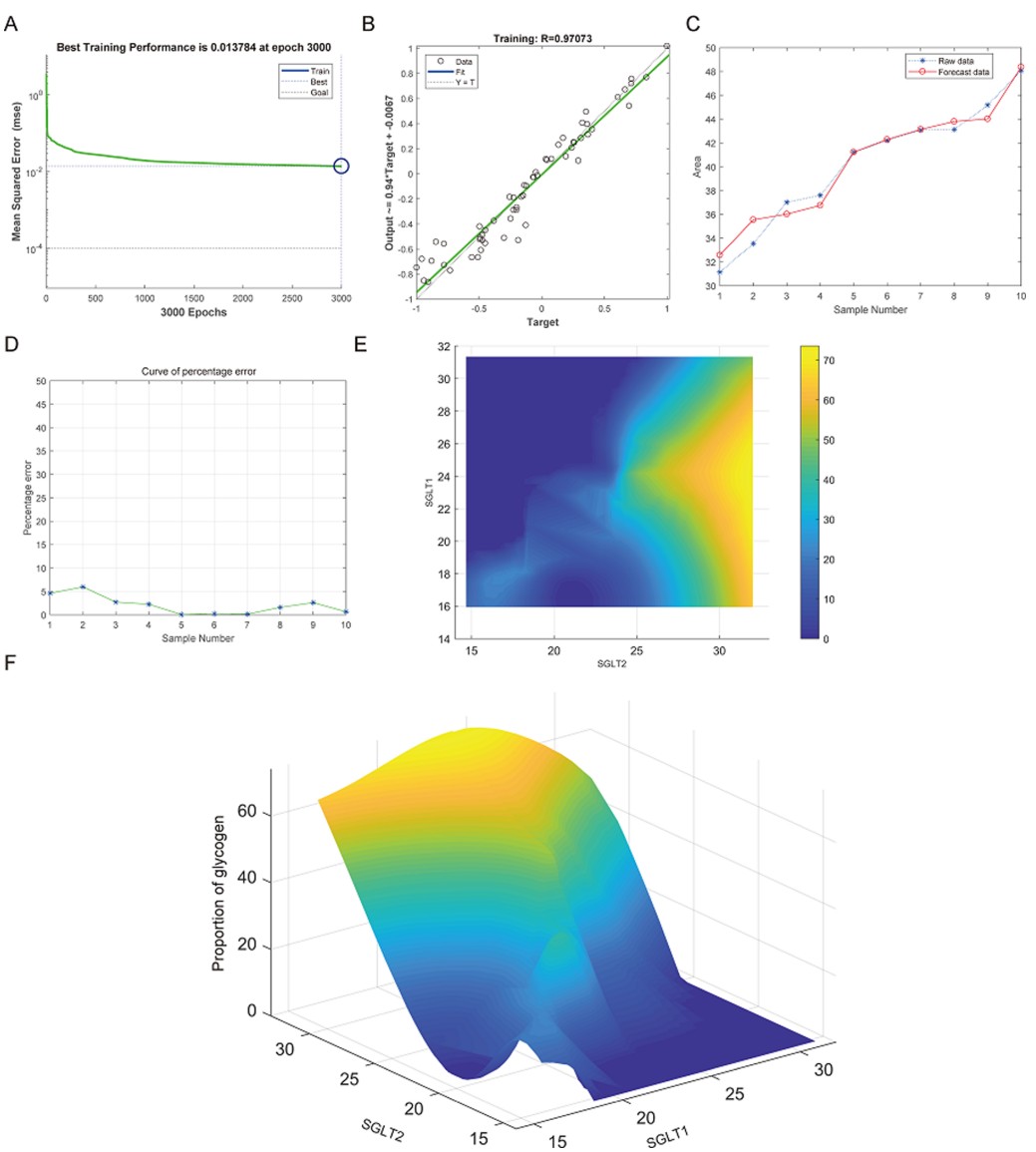

**Figure 13 Prediction and high-risk warning range of adrenal glycogen content based by SGLT1 and SGLT2 expression levels.** (A–B) The best training score was 0.013784 at epoch 3000 with a relativity of 0.97073. (C–D) Validation revealed only small differences between the predicted values and the original values. (E–F) High-risk warning indicators for adrenal gland glycogen content were determined from SGLT1 and SGLT2 expression levels, and the warning range was presented with a 3D stereogram.

with an overactive renin-angiotensin system, result in elevated homocysteine levels, varying degrees of endothelial damage. In the presence of endothelial damage, the inflammatory response drives the production of adhesion molecules, while recruitment of immune cells leads to macrophage infiltration, and eventually foam cell formation and atherosclerotic plaque formation (*Golbidi, Frisbee & Laher, 2015*). Epidemiological investigations have shown that high-intensity stress loads have become an important cause of the development of AS in modern humans (*Hintsanen et al., 2005b*; *Kivimaki et al., 2007*; *Rosvall et al., 2002*).

However, the most prominent feature during the stress response is the activation of the SNS and the HPA axis, both of which stimulate the production and secretion of hormones, including catecholamines and glucocorticoids in the adrenal glands (*Steptoe & Kivimaki, 2012*). Therefore, an appropriate mouse model of chronic stress was established to explore the effect of chronic stress on the adrenal gland. The classic indicators of chronic stress are thymus atrophy, body growth disorder, and elevated plasma glucocorticoid levels at rest. Obviously, chronic stress plays an important role in the process of cardiovascular disease. In-depth exploration of the molecular mechanism of chronic stress combined with atherosclerosis with adrenal injury is essential for the research of targeted drugs.

The adrenaline released during stress stimulates the release of glucose from the liver, which together with glucocorticoids accelerates glucose metabolism and provides sufficient energy for the body under stress. However, in the presence of prolonged stress, abnormal blood glucose levels can result from the promotion of glycogenolysis and gluconeogenesis by catecholamines, glucagon, growth hormone, glucocorticoids, and the relative deficiency of insulin (*Sharma & Singh, 2020*). Glucose is an important energy supplier in the body and is usually reabsorbed in the renal tubules, where SGLTs play an important role. SGLT1 is mainly distributed in the epithelial cells of the small intestine and is responsible for active glucose transport. It is also distributed in small amounts in the S3 region of the proximal tubule, carrying out glucose transport and reabsorption, but it is characterized by low volume and can reabsorb only 10% of glucose. SGLT2 is a high volume, low-affinity glucose transport protein located in the S1 and S2 regions of the renal proximal tubule and reabsorbs about 90% of glucose (*Wright, Ghezzi & Loo, 2017*; *Wright, Loo & Hirayama, 2011*). The current exploration of SGLTs is mainly limited to the small intestine and kidney, so we further explored the expression of SGLT1 and SGLT2 in adrenal tissues and tried to find the value and significance of SGLT1 and SGLT2 in AS models under chronic stress. In the adrenal tissue of HF+Apoe-/-+CS, the expression levels of SGLT1 and SGLT2 were found to be significantly elevated compared with the CON, CON+CS, and HF+Apoe-/- groups. Moreover, the expression levels of SGLT1 and SGLT2 in adrenal tissues were also significantly increased after chronic stress alone in non-Apoe-/- mice compared with CON mice. Specifically, adrenal SGLT1 expression was increased under chronic stress alone or AS alone, whereas AS combined with chronic stress induced greater adrenal SGLT1 expression. On the other hand, adrenal SGLT2 expression increased under chronic stress alone whereas AS combined with chronic stress promoted adrenal SGLT2 expression. The current inhibitors against SGLT2, such as dagliflozin, enagliflozin, and sotagliflozin, have cardioprotective effects and have been used in the treatment of heart failure (*Zannad et al., 2020*; *Petrie et al., 2020*; *Bhatt et al., 2021*). Perhaps, our next aim is to investigate whether SGLT2 inhibitors produce therapeutic effects on AS under dynamic chronic stress by acting on adrenal tissue and SNS.

The adrenal glands play an important role in the development of atherosclerosis; however, it is not clear what pathological changes occur in the adrenal glands in the setting of AS in combination with CS. The adrenal gland is an important endocrine gland in the body that synthesizes and secretes glucocorticoids, epinephrine and norepinephrine, and it is important to ensure normal adrenal structure and function. Therefore, we use adrenal

edema and reticular fibers to determine whether chronic stress and atherosclerosis disrupt normal adrenal tissue structure and cellular morphology, and glycogen content is related to the process of adrenal gland synthesis of various hormones. It has been found that the area of adrenal edema was significantly larger after administration of chronic stress compared to high-fat-fed Apoe-/- mice, and may be related to the increased content of secreted inflammatory factors from platelets and macrophages after over-activation of the SNS (*Cavieres, 2020*). In addition to increased cardiac output, vasoconstriction, decreased renal blood flow and sodium-water retention due to SNS hyperactivity occurs in chronic stress (*Halaris, 2013*). Our results showed that chronic stress together with AS promoted the breakage of adrenal reticular fibers and caused some disruption of adrenal cell arrangement compared to the CON+CS group. Comparing the glycogen content of the different groups, we found that chronic stress alone increased the glycogen content of adrenal tissue in the control group, and similarly, chronic stress intervention significantly increased the glycogen content of adrenal tissue in the atherosclerotic mice, which may be related to altered energy metabolism. We then focused on finding the relationship between adrenal SGLT1 and SGLT2 expression levels and adrenal injury using ROC curves, SVM and BP neural network models. It was found that all three methods consistently obtained SGLT1/2 expression levels to predict adrenal edema, reticulocyte fibrillation and abnormal glycogen content, and the fit between the predicted and actual curves gradually increased with increasing sample size with strong sensitivity and specificity. Both the BP neural network model and SVM are data-driven learning machines that can be used for predictive analysis, so the two algorithms were chosen to validate each other, circumvent the limitations of a single algorithm, and circumvent algorithmic errors.

There are some limitations in this study. The specific zonal expression of SGLT1 and SGLT2 molecules in the adrenal gland remains unclear. Therefore, we will focus on this aspect in the next research. In conclusion, to clarify the relationship between SGLT1 and SGLT2 expression levels and the extent of AS and adrenal injury in atherosclerotic mice under chronic stress intervention, we calculated the sensitivity, specificity, predictive value and high-risk warning of SGLT1 and SGLT2 expression that could be used as potential injury markers by means of ROC, SVM and BP neural network models. Therefore, SGLT1 and SGLT2 may be the molecular targets of AS and adrenal injury induced by atherosclerosis combined with chronic stress.

## CONCLUSIONS

Atherosclerosis due to chronic stress causes adrenal injury, further promoting the expression levels of adrenal SGLT1 and SGLT2 and enhancing energy metabolism. SGLT1 and SGLT2 may be molecular targets for the development of AS formation and adrenal injury in Apoe-/- mice under chronic stress.

## ACKNOWLEDGEMENTS

We thank Li Wang (Beijing Hospital) for his assistance in the animal experiment.

### Funding

The present study was funded by the National Key R&D Program of China (Grant no. 2020YFC2003000, 2020YFC2003001), the Chinese Academy of Medical Sciences, CAMS Innovation Fund for Medical Sciences (Grant no. 2018-I2M-1-002), and the National Natural Science Foundation of China (Grant no. 31271097 and 51672030). The funders had no role in study design, data collection and analysis, decision to publish, or preparation of the manuscript.

### Grant Disclosures

The following grant information was disclosed by the authors:
The National Key R&D Program of China: 2020YFC2003000, 2020YFC2003001.
the Chinese Academy of Medical Sciences.
CAMS Innovation Fund for Medical Sciences: 2018-I2M-1-002.
National Natural Science Foundation of China: 31271097, 51672030.

### Competing Interests

The authors declare there are no competing interests.

### Author Contributions

- Jianyi Li conceived and designed the experiments, authored or reviewed drafts of the article, and approved the final draft.
- Lingbing Meng analyzed the data, prepared figures and/or tables, authored or reviewed drafts of the article, and approved the final draft.
- Dishan Wu performed the experiments, prepared figures and/or tables, and approved the final draft.
- Hongxuan Xu performed the experiments, prepared figures and/or tables, and approved the final draft.
- Xing Hu performed the experiments, prepared figures and/or tables, and approved the final draft.
- Gaifeng Hu performed the experiments, analyzed the data, prepared figures and/or tables, and approved the final draft.
- Yuhui Chen performed the experiments, prepared figures and/or tables, and approved the final draft.
- Jiapei Xu performed the experiments, prepared figures and/or tables, and approved the final draft.
- Tao Gong conceived and designed the experiments, authored or reviewed drafts of the article, and approved the final draft.
- Deping Liu conceived and designed the experiments, authored or reviewed drafts of the article, and approved the final draft.

### Data Availability

The raw data and code are available in the Supplemental Files.

## Supplemental Information

Supplemental information for this article can be found online at http://dx.doi.org/10.7717/peerj.15647#supplemental-information.

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
