# Peer review of "Adrenal SGLT1 or SGLT2 as predictors of atherosclerosis under chronic stress based on a computer algorithm"

_PeerJ, doi:10.7717/peerj.15647_

## Round 0.1 · original submission · Major Revisions

Please address the concerns of all reviewers and amend the manuscript accordingly.

Reviewer 1 ·

Basic reporting

no comment

Experimental design

no comment

Validity of the findings

no comment

Annotated reviews are not available for download in order to protect the identity of reviewers who chose to remain anonymous.

Reviewer 2 ·

Basic reporting

The authors have in general provided professional figures, but I have following suggestions and comments:
(1) Figures 1 and 2 seem to be of low resolution. In particular, a larger font should be used for the y-axis labels.
(2) Figure 8 captions for A and E both refer to the intima-media thickness of abdominal aorta when the caption for E should refer to the internal diameter of the abdominal aorta.
(3) The y-axis labels for the first panels of Figure 10B,C are mislabeled.
(4) The titles of Figure 9A all read SGLT1. They would be better understood if changed to include the properties SGLT1 was used to predict, namely, adrenal edema, broken reticular fibers, and glycogen content. The same applies to Figure 9B titles.
(5) Figure 3A: do the authors mean to show scale on the right-hand side of the figure?

Experimental design

Li and Meng et al. conducted systematic experiments to study the physiological effects of atherosclerosis under chronic stress using an appropriate mouse model. They observed the pathological changes of the aorta and adrenal glands using histological methods. Using computer algorithms such as ROC curves, SVM, and neural network, the authors show that the elevated expression levels of SGLT1 and SGLT2 can be used to predict adrenal abnormalities.

Validity of the findings

(1) The authors state on lines 281-282 that the HF+Apoe-/-+CS group had statistical difference in quiescent time in closed arms from the HF+Apoe-/- group after 12 weeks of CUMS. However, no such statistical difference is indicated on Figure 1D.
(2) Similarly, the authors state on lines 290-291 that after 12 weeks, the total quiescent time in the CON+CS was greater than that in the CON group. This is not consistent with Figure 2D.
(3) According to the authors, that mice with prolonged quiescent time at total distance in the elevated plus maze test indicated anxious state (lines 164-165). However, the experiment does not reflect this (the first column of panels in Figure 1). In fact, after 12 weeks, the experiment showed the opposite with statistical significance (CON+CS vs. CON and HF+Apoe-/-+CS vs. HF+Apoe-/-). What accounts for this discrepancy?
(4) The authors state on lines 427-429 that SGLT1 expression can be increased with atherosclerosis alone, whereas the expression of SGLT2 was down-regulated in AS alone. What are the data that support this observation? I could not find meaningful difference between SGLT1 and SGLT2 expression levels from Figures 5 and 7. The authors also repeat this distinction on lines 482-485.
(5) What are the cutoffs for the risk of adrenal edema, reticular fiber rupture, and glycogen content? And what are the rationals for choosing such cutoffs? This is critically important for defining the appropriate ranges of SGLT1/2 expression levels, yet they are not addressed in the manuscript.
(6) The authors identified 3 different ranges of SGLT1/2 expression levels that can increase the risk of adrenal edema, reticular fiber rupture, and glycogen content. What conclusion can we draw from these? A consensus region or any other way to synthesize the results would be helpful.
(7) The Conclusion section should be stated more fully. The authors begin stating conclusions on line 512 in the Discussion section. Some re-organization is needed.

Additional comments

(1) Please explain the term SEM on line 257. Is it Standard Error of the Mean?

Reviewer 3 ·

Basic reporting

In This work, Jianyi and colleagues have presented the data to provide information regarding distribution and the expression levels of SGLT1 and SGLT2 in adrenal gland during pathological changes in chronic stress and the development of AS. The most important finding of the study is the Ultrasonography visualization findings are supported with HE staining data of the aorta indicating the larger size plaque formation in HF+Apoe-/-+CS group mice. In the discussion the authors elaborated the research findings with the valid references. I found this article is very important for further studies to diagnose or identify Atherosclerosis (AS) at the earliest possible using Sodium transporters as the biomarkers for the indication.
The Explanation for each research finding was very clear
The Experimental design is very clear and valid to support the study.

Experimental design

The Explanation for each research finding was very clear
The Experimental design is very clear and valid to support the study.

Validity of the findings

Lane 130: Chronic unpredictable mild stress (CUMS) was written as CMUS. Please Correct it.
Figure 5a-b: In the explanation about the data, authors mentioned about the highest. This highest is compared with zero week staining or anything else, it’s not clear.
Lane 493: As mentioned, Increased secreted inflammatory factors from platelets and macrophages, Did authors measured any inflammatory genes or markers??
As Suggestion: Authors can also perform the q-PCR of SGLT1 and SGLT2, Which will the great supporting for the findings with Staining (Figure 5: Semi-quantitative).

Annotated reviews are not available for download in order to protect the identity of reviewers who chose to remain anonymous.

---

## Round 0.2 · accepted · Accept

All issues pointed out by the reviewers were adequately addressed and the manuscript was amended accordingly. Therefore, I am pleased to inform you that the revised version is acceptable now.

Reviewer 2 ·

Basic reporting

The authors have satisfactorily addressed my comments.

Experimental design

The authors have satisfactorily addressed my comments.

Validity of the findings

The authors have satisfactorily addressed my comments.